# Optimal compressed representation of high throughput sequence data via light assembly

Antonio A. Ginart[1], Joseph Hui[2], Kaiyuan Zhu[3], Ibrahim Numanagić[4], Thomas A. Courtade[5] S. Cenk Sahinalp[3] & David N. Tse[1]

The most effective genomic data compression methods either assemble reads into contigs, or replace them with their alignment positions on a reference genome. Such methods require significant computational resources, but faster alternatives that avoid using explicit or de novo-constructed references fail to match their performance. Here, we introduce a new reference-free compressed representation for genomic data based on light de novo assembly of reads, where each read is represented as a node in a (compact) trie. We show how to efficiently build such tries to compactly represent reads and demonstrate that among all methods using this representation (including all de novo assembly based methods), our method achieves the shortest possible output. We also provide an lower bound on the compression rate achievable on uniformly sampled genomic read data, which is approximated by our method well. Our method significantly improves the compression performance of alternatives without compromising speed.

[1] Department of Electrical Engineering, Stanford University, Stanford, CA 94305, USA. [2] Department of Electrical Engineering & Computer Science, Massachusetts Institute of Technology, Cambridge, MA 02139, USA. [3] Department of Computer Science, Indiana University Bloomington, Bloomington, IN 47405, USA. [4] Computer Science & Artificial Intelligence Laboratory, Massachusetts Institute of Technology, Cambridge, MA 02139, USA. [5] Department of Electrical Engineering and Computer Sciences, University of California, Berkeley, CA 94720, USA. Antonio A. Ginart, Joseph Hui and Kaiyuan Zhu contributed equally to this work. Correspondence and requests for materials should be addressed to K.Z. (email: kzhu@indiana.edu) or to S.C.S. (email: cenksahi@indiana.edu)

Recent advances in high-throughput sequencing (HTS) have made it possible to sequence genomes of complex organisms in a matter of hours. As worldwide increase in genomic sequence data generation put strain on storage and communication systems, new compression methods are designed to reduce this burden by exploiting redundancy within and across reads extracted from genome sequences. Currently, available HTS data compression methods reduce the redundancy within a genomic sequence data set by either (i) assembling reads into long contigs, typically by a de Bruijn graph-based approach (e.g., Quip[1], Leon[2], k-Path[3], and KIC[4]), or by (ii) aligning the reads to a reference genome (e.g., LW-FQZip[5]); the reads are then encoded as simple pointers to the reference or the assembled contigs.

Because both sequence mapping and assembly are computationally intensive tasks, all the above HTS compression methods are typically slow and thus are not commonly used even though they achieve high compression performance. The best compromise between the compression rate and running time is typically achieved by HTS data compressors that perform read mapping or assembly only implicitly; these methods first cluster reads that share long substrings, and independently compress reads in each cluster after a reordering or implicit assembly. Since the read order within unmapped read files (in FASTA/Q formats) is not important, reordering of the reads in a manner that brings the similar reads together can significantly boost the compression rates while avoiding information loss (e.g., SCALCE[6], Orcom[7], Mince[8], and BEETL[9]). If the underlying genome is highly repetitive, or if the coverage of the data is high, even general-purpose compressors, such as gzip[10] or bzip[11], significantly benefit from the improvement in data locality as a result of reordering.

In this paper, we introduce a new reference-free compressed representation for HTS data that is based on a crude yet more explicit de novo assembly of reads for improving compression rates. (Note that compression performance is measured in terms of either "compression rate", the number of bits in the output per each input bit, or "compression ratio", its inverse.) Our representation represents a set of input reads as nodes of compact tries. Each edge, from a node $v$ to its parent $u$, represents the suffix of the read corresponding to $v$ that is not covered by the read corresponding to $u$. Unique to our representation, the tries are not organized in a top-down, but rather in a bottom-up fashion, i.e., each node $v$ has a link only to its parent $u$ and not to its children.

Next, we describe an iterative method called Assembltrie, to build a forest of such tries, which, for each read $r$, (i) greedily picks an already processed read $r'$ for which the overlap between a prefix of $r$ and the suffix of $r'$ is maximum possible, and a new node $v$ is created with a link to the node $u$ corresponding to $r'$ as its parent (if no such $r'$ exists, it starts a new tree with $r$ only), and, (ii) greedily identifies each (already processed) read $r''$, which has a longer prefix that match a suffix of $r$ than that of the read corresponding to its existing parent—and updates its parent as $v$.

As a first in HTS data compression, we show that Assembltrie achieves optimality from a combinatorial point of view (on finite size HTS data) as follows. Given a finite HTS read collection to be compressed by an explicit or implicit representation of the input as a subgraph of the standard read overlap graph, where reads are represented through pointers to nodes, Assembltrie is guaranteed to produce the smallest possible representation in terms of the number of symbols used in the subgraph as well as the number of pointers; Assembltrie uses no additional information to maintain the topology of the subgraph. This guarantee covers all algorithms that are constrained with the compressed representation described above, including those methods that represent the reads as pointers to their (string graph based) de novo assembly.

Note that this notion of combinatorial optimality is achieved for read sets that do not have read errors; it also does not extend to compressors that represent the input as a subgraph of a de Brujn graph (even though in such representations, the number of pointers could be superlinear with the number of reads). Furthermore, our notion of optimality does not imply bitwise optimality on all inputs due to standard information theoretic limitations.

We also provide the first information theoretic lower bound on the compression rate achievable on a collection of reads that are uniformly selected from a reference with known entropy, and demonstrate that Assembltrie closely approaches this ultimate performance limit.

We have evaluated our method on a recent benchmarking data set[12] comprised of unmapped read (FASTQ) collections with both deep and shallow coverage from various organisms (*H. sapiens*, bacteria, plants) and from several sample types (genomic and metagenomic), as well as some simulated and aligned (i.e., reads are extracted SAM/BAM files, where the positions of the reads were discarded) HTS data. Our method improves the compression rate achieved by the best available software tools, such as Orcom[7] or Mince[8], by 10–50%, depending on the type of the data set. With respect to the running time, Assembltrie is competitive with the reordering-based methods and offers remarkable gain over the alignment/assembly based methods. Furthermore, the runtime performance is significantly improved through parallelization without sacrificing compression performance.

## Results

Assembltrie was tested on a Linux server equipped with 39 10-core Intel® Xeon® E5-2650 v3 2.30 GHz CPUs, 1058 GB of RAM in total, and a Lustre-based file system with 750TB disk space. For convenience, the units in this section take decimal multiples, i.e., $1 M = 10^6$, $1 G = 10^9$ etc., to represent the number of reads, file sizes, and memory usage.

**Benchmarking Assembltrie against other methods**. The data sets we used to benchmark Assembltrie consists of a number of FASTQ files compiled by MPEG HTS working group, composed of ~200 GB of human, human microbiome (metagenomic), bacterial and plant genome sequence, and gives a comprehensive assessment for the relative performance of Assembltrie against all existing FASTQ compression methods. (See Table 1.) Note that since the read collection from the *T. cacao* genome are of poor quality with a very high error rate, we also added a simulated set

---

**Table 1 The MPEG HTS benchmarking data set**

| Sample | Organism | Genome Len. (M) | # Reads (M) | Read Len. |
|---|---|---|---|---|
| SRR554369 | *P. aeruginosa* | 6.60 | 1.66 | 100 |
| SRR327342 | *S. cerevisiae* | 12.14 | 15.04 | 63 |
| MH0001.081026 | *H. sapiens gut* | N/A | 11.64 | 44 |
| SRR870667 | *T. cacao* | 335.44 | 69.10 | 108 |
| ERR174310 | *H. sapiens* | 2989.43 | 207.58 | 101 |
| ERP001775 | *H. sapiens* | 2989.43 | 607.56 | 101 |
| Simulated | *T. cacao* | 335.44 | 65.43 | 108 |
| NA12878 | *H. sapiens* | 2989.43 | 226.11 | 101 |

The HTS read collections used to evaluate the performance of Assembltrie from the MPEG HTS benchmarking data set. Note that ERP001775 is a large data set, which combines reads from 18 human individuals. It was downsampled to fit the memory requirements of Assembltrie. In addition, NA12878 is not part of the MPEG HTS FASTQ/FASTA benchmarking data set—reads in this data set have been extracted from a corresponding BAM file to demonstrate the comparative performance of Assembltrie and Orcom on data, where strand correction is not needed. Finally, we used a simulated *T. cacao* data set instead of the original due to the high error rate observed in the original data set

**Table 2 Comparative compression ratios achieved by Assembltrie on the MPEG HTS benchmarking data set**

| Sample | L / cov | Compression rates | | | | | |
|---|---|---|---|---|---|---|---|
| | | Assembltrie | Assembltrie (corrected) | Orcom | BEETL | Mince | k-Path |
| SRR554369 | 100/25 | 0.369 | 0.345 | 0.518 | 1.133 | 0.484 | 0.673 |
| SRR327342 | 63/80 | 0.272 | 0.291 | 0.304 | 0.986 | 0.312 | 0.384 |
| MH0001.081026 | 44/NA | 0.781 | 0.758 | 0.804 | 1.785 | 0.786 | 2.545 |
| SRR870667 | 108/20 | 1.821 | 1.733 | 0.884 | 1.287 | 0.735 | 0.707 |
| ERR174310 | 101/7 | 0.701 | 0.570 | 0.686 | 1.493 | 0.746 | 0.797 |
| ERP001775 | 101/20 | 0.350 | 0.322 | 0.364 | N/A | N/A | N/A |
| Sim. *T. cacao* | 108/19 | 0.538 | 0.479 | 0.667 | N/A | N/A | N/A |
| NA12878 | 101/7 | 0.444 | N/A | 0.650 | N/A | N/A | N/A |

Compression rates in bits per base for each software tool (with 8 threads) and each MPEG benchmark sample. The second column provides the read length and coverage; and the last columns present the compression performances for different software tools. Assembltrie outperforms all of the existing sequence-only compressors with different level of improvement depending on the read length/coverage (possibly with the greedy strand correction heuristic), except on the sample SRR870667 from *T.cacao* (which has an unusually high error rate)

**Table 3 Compression times and memory usage of Assembltrie and Orcom**

| Sample | Assembltrie (time) | | Assembltrie (RAM) | | Orcom | Orcom |
|---|---|---|---|---|---|---|
| | without / with strand correction | | without / with strand correction | | (time) | (RAM) |
| SRR554369 | 23.2 | 30.6 | 1275.9 | 1372.8 | 10.1 | 631.0 |
| SRR327342 | 256.2 | 439.9 | 8553.6 | 7971.8 | 131.0 | 1767.8 |
| MH0001.081026 | 118.5 | 145.7 | 7662.4 | 7223.5 | 49.6 | 1674.2 |
| SRR870667 | 15251.1 | 12219.2 | 50580.9 | 53184.3 | 919.0 | 3150.6 |
| ERR174310 | 31657.8 | 43758.2 | 152719.8 | 168861.1 | 6411.1 | 3439.2 |
| ERP001775ª | 22227.0 | 31675.1 | 434425.5 | 473066.5 | 8969.1 | 10803.6 |

Compression time (in seconds) and memory usage (in MBs) of Assembltrie and Orcom to generate the compression rates for the MPEG benchmark data set. The results of Assembltrie are given both with (the 3rd and 5th columns) and without (the 2nd and 4th columns) heuristic strand correction
ªAssembltrie was tested with non-default parameters

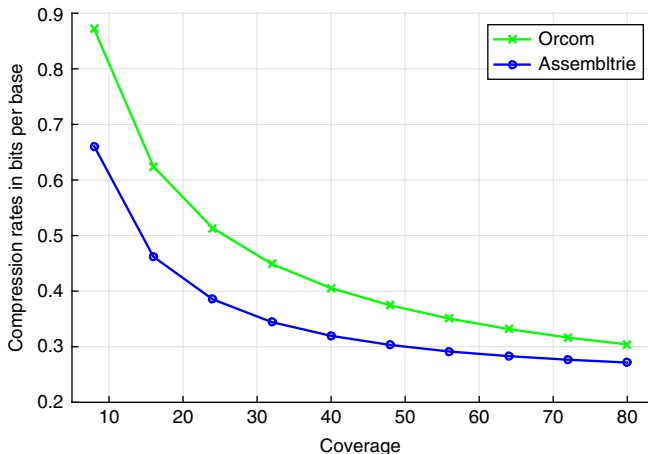

**Fig. 1** Compression performance of Assembltrie as a function of read coverage. The compression performance (with 8 threads) of Assembltrie in comparison to Orcom on downsampled read collections from SRR327342 —as a function of coverage

unmapped reads in FASTQ/FASTA format, and a conversion from mapped reads to unmapped read collections is wasteful. However, we used this data set to measure the effectiveness of the strand correction heuristic used by Assembltrie (see "Methods" section) since reads in a SAM/BAM file are already strand corrected.

**Compression performance and computational resource utilization.** The experimental results, including the compression performance, and the computational resources used by our method are presented in Tables 2 and 3. (Note that compression performance is measured in terms of either "compression rate", the number of bits in the output per each input bit, or "compression ratio", its inverse.) In addition to Assembltrie, we present the compression performance (on the MPEG HTS compression benchmark data set) of Orcom[7], BEETL[9], Mince[8], and k-Path[3], as provided in the MPEG benchmarking study[12]. As per Assembltrie, all of these compressors are developed to compress only the sequence information in a FASTQ file. Among these methods, we observed that Orcom is much faster than the others thus we compared the compression time of Assembltrie against Orcom only.

Table 2 presents the compression rates achieved by the above methods on the MPEG benchmark data set. As we can see, Assembltrie outperforms alternatives on all samples, typically providing an improvement of 10–35% in compression ratio in comparison to the best alternative (usually Orcom or Mince)— with the exception of the SRR870667 data set. This data set from the plant *T. cacao* is comprised of low-quality reads: in fact, more than 40% of the reads do not share an overlap greater than $\lfloor L/5 \rfloor$ = 21 with any other read (here $L$ = 108), even when the number of mismatches allowed is 4. In contrast, more than 90% of the reads

of reads sampled from the *T. cacao* reference genome (with a uniform i.i.d. error rate of 1%) for comparison. In addition to this data set collection, we have extracted reads from a BAM-formatted human genome data set NA12878 and converted them to FASTA format. (This file is not a part of the MPEG HTS working group FASTQ/FASTA compression benchmarking data set. Instead, a higher coverage version is a part of the SAM/BAM benchmarking data set. We downsampled it to match the coverage of ERR1743010.) Assembltrie is developed for compressing

**Table 4 Decompression times and memory usage of Assembltrie and Orcom**

| Sample | Assembltrie (time) | | Assembltrie (RAM) | | Orcom | Orcom |
|---|---|---|---|---|---|---|
| | without / with strand correction | | without / with strand correction | | (time) | (RAM) |
| SRR554369 | 2.9 | 3.4 | 249.7 | 259.0 | 4.7 | 15.2 |
| SRR327342 | 18.0 | 36.1 | 1558.7 | 1678.9 | 30.8 | 30.2 |
| MH0001.081026 | 13.7 | 15.3 | 1032.4 | 1133.7 | 24.9 | 23.7 |
| SRR870667 | 190.3 | 207.5 | 10500.8 | 11495.9 | 673.6 | 708.6 |
| ERR174310 | 524.1 | 775.8 | 31571.0 | 33555.2 | 707.3 | 561.7 |
| ERP001775 | 758.7 | 860.3 | 86468.1 | 91465.8 | 1487.9 | 1532.5 |

Decompression time (in seconds) and memory usage (in MBs) of Assembltrie and Orcom in single-threaded mode. The results of Assembltrie are given both with (the 3rd and 5th columns) and without (the 2nd and 4th columns) heuristic strand correction

**Table 5 Assembltrie's compression performance is comparable to our entropy approximation for _E. coli_ read collection**

| Sample | L / cov | LZ(G) | $H(R^*\|G)$ | $H(\mathcal{R}\|\mathcal{R}^*)$ | $H(\mathcal{R})$ | Assembltrie | Orcom |
|---|---|---|---|---|---|---|---|
| DH10B 1 | 120/40 | 0.048 | 0.020 | 0.047 | 0.115 | 0.146 | 0.372 |
| DH10B 2 | 100/40 | 0.048 | 0.025 | 0.053 | 0.126 | 0.163 | 0.399 |
| DH10B 3 | 80/40 | 0.048 | 0.028 | 0.042 | 0.119 | 0.164 | 0.379 |
| DH10B 4 | 100/25 | 0.077 | 0.031 | 0.053 | 0.162 | 0.194 | 0.507 |
| DH10B 5 | 100/80 | 0.024 | 0.017 | 0.053 | 0.094 | 0.141 | 0.292 |
| DH10B 6 | 100/40 | 0.048 | 0.025 | 0.018 | 0.092 | 0.123 | 0.290 |

The entropy approximation of the reads from the above _E. coli_ read collections. LZ(G) denotes the bits/bits per base after compressing each genome with gzip; $H(R^*|G)$ is the entropy approximation based on a multinomial sampling of each reference genome; $H(\mathcal{R}|\mathcal{R}^*)$ is the binary entropy of the error process; and $H(\mathcal{R})$ is the overall entropy approximation for each read collection. Finally, we compare the compression results given by Assembltrie (run with a single thread) and Orcom

from the simulated _T. cacao_ data (with similar coverage and depth, but with 1% error rate) get "assembled" (placed in the cycle-rooted tries).

Note that Assembltrie performs especially well on the low-coverage samples. In fact, as illustrated in Fig. 1, the compression gap between Assembltrie and Orcom increases as the coverage decreases. Table 2 also demonstrates the effectiveness of our greedy strand correction heuristic (see "Methods" section for details). As can be seen, Assembltrie can achieve as much as 20% gain on the compression ratio by applying a simple greedy strand correction heuristic (which works especially well on low-coverage data); however, the heuristic cannot determine the correct strand orientation for many of the reads. To demonstrate the gap between what our greedy heuristic achieves and how Assembltrie performs on data with reads that are all correctly oriented, we give results on a read collection we extracted from a human genome BAM file, NA12878—where reads are already strand corrected. We extracted reads from this BAM file while ensuring that the resulting data set would have coverage and read length similar to that of ERR174310. The two human data sets are compressed equally well by Orcom. Interestingly, there is a big gap between the compression rates achieved by Assembltrie on these files. Orcom's performance is comparable to Assembltrie on the non-strand-corrected file, when the heuristic is not used. The use of the heuristic improves Assembltrie results by ~20%. On top of this, Assembltrie achieves another 22% improvement on data that is already strand corrected. In other words, Assembltrie is likely to provide significantly improved compression rates with better strand correction methods. (It is possible to perform mapping for strand correction but that would add burden on the running time.)

Table 3 presents the running time and memory usage of Assembltrie in comparison to Orcom. Note that Assembltrie is slower than Orcom with default read overlap length $K = \lfloor L/5 \rfloor$ but with increasing $K$ it achieves a similar running time (with the exception of the single problematic read collection—SRR870667,

where lower compressibility results in a higher search time for the placement of each read). Further, note that the runtime and RAM usage of Assembltrie is affected by the use of its strand correction heuristic. In terms of memory usage, Orcom has an advantage since Assembltrie must maintain all of the processed reads as well as the prefix and suffix hash tables, necessitating $\Omega(NL)$ memory.

Finally, Table 4 presents the running time and memory usage of Assembltrie vs Orcom for decompression purposes. Assembltrie is consistently faster than Orcom, with or without strand correction, sometimes significantly so (e.g., for SRR870667 data set, Assembltrie decompression time is 3.5 times faster). The memory usage of Orcom, however, is better—as explained above.

**Assembltrie performance vs our entropy approximation**. We have also compared Assembltrie's performance with our entropy approximation for a given collection of HTS reads (see "Methods" section for details). For this, we generated six simulated data sets, where the reads were obtained by randomly sampling _E.coli_ K-12 DH10B genome (length 4.69 Mbases), varying the read length, coverage, and error rate (the probability that each base is mutated). The number of reads (in millions), read length, coverage, and error rate in each of these data sets are (respectively): DH10B 1: 1.56 M, 120 bp, 40x, 0.30%; DH10B 2: 1.88 M, 100 bp, 40x, 0.35%; DH10B 3: 2.34 M, 80 bp, 40x, 0.27%; DH10B 4: 1.17 M, 100 bp, 25x, 0.35%; DH10B 5: 3.75 M, 100 bp, 80x, 0.35%; and DH10B 6: 1.88 M, 100 bp, 40x, 0.10%.

Table 5 presents the performance of Assembltrie on these simulated data sets. The entropy rate (in bits per base) $H(\mathcal{R})$ of each read collection $\mathcal{R}$ is calculated according to Eq. (1) (see "Methods" section). Clearly, Assembltrie is the only software tool we tested that produces compression results that come close to $H(\mathcal{R})$; there is a big gap between our entropy approximation and even the best performing Orcom's compression results. In fact, Assembltrie outperforms Orcom by a factor of 2.1 to 2.6 in these data sets.

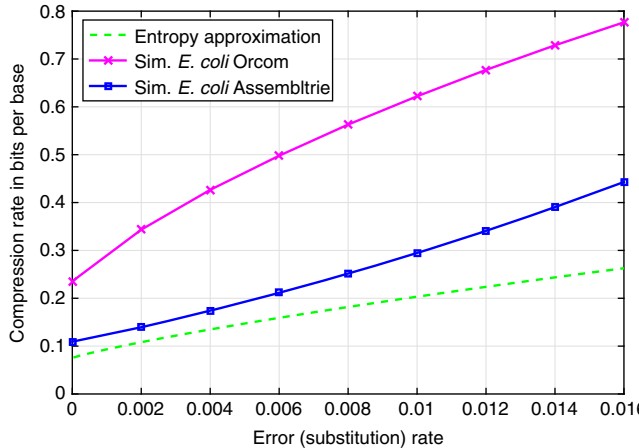

**Fig. 2** Compression performance of Assembltrie as a function of error rate. The compression ratio achieved by Assembltrie is close to our information theoretic approximation on simulated reads from E.coli K-12 DH10B genome (the data set involves 1.8 M "reads," i.e., substrings of length $L = 101$, sampled uniformly with simulated errors)

In addition to the data sets above, we have also experimented with simulated read collections sampled from the same *E.coli* strain, this time with fixed coverage (1.8 M reads all together) and read length (101 bp) but with varying error rate. Figure 2 demonstrates how Assembltrie's performance varies on this data set as a function of the error rate (here only substitutions are considered as errors since they are the major form of sequencing errors in Illumina data). As can be expected, the gap (shown in Fig. 2) between the maximum achievable compression ratio and what is obtained by Assembltrie grows as the error rate increases. This gap is primarily due to Assembltrie's requirement of an exact $K$-mer match during the process of identifying potential parents/ children for each read. As the error rate approaches 0, the gap between Assembltrie's performance and our entropy approximation diminishes.

**Parameter selection**. As mentioned earlier, the running time of Assembltrie is primarily determined by the minimum overlap length $K$—the larger value of $K$ results in a shorter running time. Unfortunately, larger $K$ values may result in missed overlaps, potentially sacrificing overall compression performance. Figure 3 depicts the tradeoff between Assembltrie's running time and compression performance, with varying values of $K$. On read data sets from a large (e.g., human) genome, choosing $K \geq \lfloor L/4 \rfloor$ provides a good tradeoff between the running time and compression performance.

**Discussion**

As demonstrated above, Assembltrie is a high-performance sequence compression tool for large genomic read collections, capable of improving the compression ratio achieved by all available methods significantly. It may be possible to further improve its performance through improved strand correction heuristics; however, this should not come at the expense of a poorer running time. Assembltrie's main contribution is in providing improved compression without sacrificing running time. However, its memory usage is relatively high. It may be possible to reduce the memory usage by providing a tradeoff between running time or compression ratio and the memory usage by limiting the branching factor in the compact tries it builds (in fact limiting the branching factor to 1 will result in contigs that will not only avoid the use of pointers and improve the memory

usage, but also improve the running time—however, this is likely to result in poorer compression performance).

**Methods**

**Problem definition**. The main goal of Assembltrie is to achieve lossless compression of a collection (multiset) of fixed length genomic reads, i.e., strings from the four letter DNA alphabet (and possibly the letter N—to represent unknown nucleotides), denoted by $\mathcal{R}$ (see the literature[13–18] for discussions on the general problem of lossless multiset compression). Since the reads form a multiset, and because the read locations are arbitrary, we do not maintain the order of the reads.

Assembltrie is based on a light assembly of reads in the sense that instead of assembling the reads into independent contigs, they are organized in a more compact data structure that we call a read forest. Specifically, a read forest $\mathcal{T} = (V, E, w, \text{st})$ is a directed graph, where each node $v \in V$ corresponds to a specific read $r \in \mathcal{R}$ associated with a string $\text{st}(v)$ of fixed length $L = |\text{st}(v)|$. Each connected component of $\mathcal{T}$ can be thought as a trie-like connected subgraph $T = (V_T, E_T, w, \text{st})$ ($V_T \subseteq V, E_T \subseteq E$), such that for each node $v$ in $V_T$, there is exactly one outgoing-directed edge, say $(v, u)$, where $u \in V_T$ is said to be the parent of $v$ and is denoted $\pi(v) = u$ (it is possible that $\pi(v) = \text{NIL}$ indicating that $v$ is a root). Each edge $(v, u) \in E_T$ is said to have weight $w(v, u)$, which is the length of the shortest suffix of $\text{st}(u)$ that cannot be covered (i.e., matched exactly) by a suffix of $\text{st}(u)$. In other words, $w(v, u) = L - l$, implies that $\text{st}(v)[1:l] = \text{st}(u)[L - l + 1:L]$, where $l$ is the length of such suffix–prefix overlap between $\text{st}(v)$ and $\text{st}(u)$, and $s[i:j]$ denotes the substring from index $i$ to $j$ in string s. Since $T$ is comprised of nodes with a single outgoing edge, it forms a graph with at most one cycle. Note that each edge of $T$ represents a string; hence, we can think of $T$ as a trie, which has either a single node or a cycle (instead of a single node) acting as the root. As a result, the set $\mathcal{T}$ of trie-like graphs $T$ (we will call them cycle-rooted tries, with an example given in Fig. 4) forms a forest, which can be thought of as a subgraph of the standard read-overlap graph (one of the two basic frameworks commonly used in genome assembly—see ref. [19] for a definition) formed by the input reads.

Unique to our representation, a cycle-rooted trie $T$ is not arranged in a top-down, but rather in a bottom-up fashion, since each node $v$ has a link only to its parent $\pi(v)$. Thus, the construction of a read forest $\mathcal{T}$ simply constitutes the computation of the parent node $\pi(v)$ for each node $v$. Since the compressed representation of the input $\mathcal{R}$ is simply an efficient encoding of $\mathcal{T}$, the objective of Assembltrie is to construct a read forest $\mathcal{T}^*$ that contains minimum number of symbols, i.e., in which the sum of edge weights is minimum possible.

Given a read forest $\mathcal{T}$, let its *total weight* be the sum of the weights of its edges, i.e.,

$$|\mathcal{T}| = \sum_{T \in \mathcal{T}} \sum_{v \in V_T} w(v, \pi(v)),$$

where we can assume that $w(r, \text{NIL}) = L$ for each root node $r$. In the remainder of the paper, we will consider read forests $\mathcal{T}_K$, where the minimum overlap length between any string $\text{st}(v)$ and $\text{st}(\pi(v))$ is a user specified $K$ (see Fig. 3 for details). As we will show, Assembltrie produces the read forest with minimum total weight, i.e., $\mathcal{T}^* = \text{argmin}|\mathcal{T}_K|$, for any value of $K$. Compared with the shortest superstring $\mathcal{S}$— as an alternative representation of the input set $\mathcal{R}$ of reads—$\mathcal{T}^*$ is guaranteed to contain at most as many symbols (and possibly less), while maintaining a single pointer per read.

**Constructing read forest**. As mentioned above, in order to construct the read forest $\mathcal{T}_K$, one simply needs to identify for each node $v$, the parent node $\pi(v)$. This process is performed iteratively: for each distinct read $r \in \mathcal{R}$ Assembltrie iteratively considers, it creates a new vertex $v$ that corresponds to $r$. (Note that the number of occurrences of each specific read can be maintained in a separate data structure.) It then greedily picks an existing node $\pi(v)$, satisfying two conditions: (i) the overlap between suffix of $\pi(v)$ and prefix of $v$ is maximal among all existing nodes and (ii) the overlap size is at least $K$. In case the overlap is shorter (than $K$), $v$ forms an independent (cyclic) trie with $\pi(v) = \text{NIL}$. Next, Assembltrie identifies all children of $v$, i.e., those nodes $u$ such that the (longest) overlap between a prefix of $\text{st}(u)$ and a suffix of $\text{st}(v)$ is longer than that between a prefix of $\text{st}(u)$ and a suffix of $\text{st}(\pi(u))$, provided that the overlap length is, again, at least $K$. For each such node $u$, its parent $\pi(u)$ is reset to $v$.

In order to find the suffix–prefix overlaps, Assembltrie constructs two hash tables, one for maintaining the length $K$ prefix and the other for the length $K$ suffix of each of the reads. Given a node $v$, Assembltrie searches its potential parents by sliding a window of length $K$ across $\text{st}(v)$ from right to left, identifying each node $u$ for which the length $K$ suffix of $\text{st}(u)$ exactly matches this window. In case this initial match extends to a full match between the prefix of $\text{st}(v)$ that includes this window, and the corresponding suffix of $\text{st}(u)$ of identical length (see Implementation Details for a detailed description of how we extend an initial match), $u$ is declared as the parent of $v$, completing the search. If no such node $u$ can act as a parent of $v$, then the window slides one position to the left and the search resumes. Assembltrie searches for possible children of each vertex $v$ in a similar manner; note that we now need to slide the window from left to right, and use the hash table for prefixes. If for a matching node $u$, $\text{st}(u)$ has an overlap with $\text{st}(v)$, which is longer than that with its current parent, i.e., $w(u, v) < w(u, \pi(u)) \leq L -$

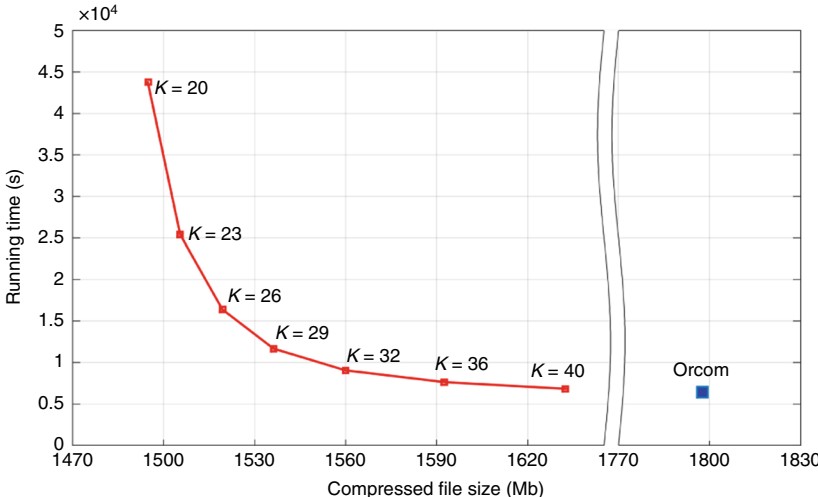

**Fig. 3** The tradeoff between the running time and compression performance of Assembltrie. The tradeoff between Assembltrie's running time and compression rate as a function of the user set parameter $K$ (initial match length) on the data set ERR174310. The blue point depicts the performance of Orcom on the same data set with default parameters

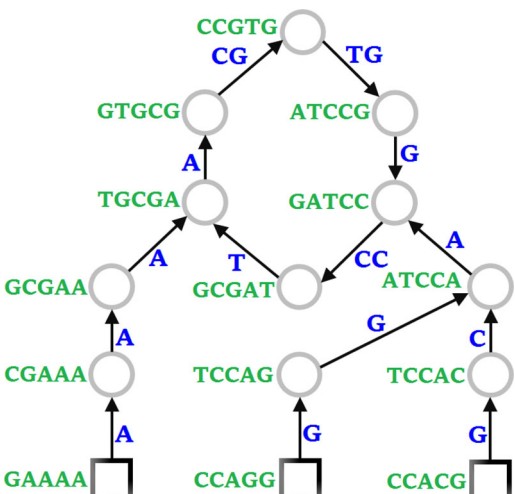

**Fig. 4** Cycle-rooted trie. An example cycle-rooted trie constructed from a collection of reads of length 5, with a cycle (in the center) constituting the root

$K$, then $\pi(u)$ is reset to $v$. After the parent and the children of $v$ are established, the suffix and prefix of $\mathrm{st}(v)$ of length $K$ are inserted into the corresponding hash tables. The overall workflow of Assembltrie is depicted in Fig. 5.

**Running time**. The process of identifying the potential parent and children of each read (or node) takes $O(m(L - K))$ time, where $m$ is the maximum load of an entry in either hash table. This implies that in the worst case we need $O(mN(L - K)) = O(mNL)$ time to construct the read forest. Under the condition that the underlying reference genome $G$ is uniform i.i.d. and the reads $\mathcal{R}$ are extracted from $G$ uniformly at random, $m$ will be a small constant[20, 21] and the expected time to construct the read forest becomes $O(NL)$. In practice, however, long genomic repeats as well as sequencing errors lead to hash collisions and hence a moderate slowdown.

**Combinatorial optimality**. We now prove that the greedy algorithm used by Assembltrie actually produces the optimal (that is, the one containing minimum total number of symbols) read forest $\mathcal{T}^*$, based on the assumption that the reads have no sequencing errors. In particular, the total weight of $\mathcal{T}^*$ ends up at most as much as the length of a shortest superstring constructed from $\{\mathrm{st}(v) : v \in V\}$, since each superstring must correspond to a set of paths that connect all $v \in V$, which also forms a valid read forest $\mathcal{T}$.

In order to show the invariant maintained through the greedy construction of $\mathcal{T}_K = (V, E_K, w, \mathrm{st})$, we extend the definition of the shortest non-overlapping

length $w$ to each pair of nodes and define for each node $v \in V$ its representation as:

$$\mathrm{rep}(v) = \min \begin{cases} w(v, u) & \text{if } w(v, u) \le L - K, u \ne v \in V \\ L & \text{otherwise} \end{cases}.$$

**Lemma 1** *For any read forest $\mathcal{T}_K = (V, E_K, w, \mathrm{st})$, its total weight* $|\mathcal{T}_K| \ge \sum_{v \in V} \mathrm{rep}(v)$.

*Proof* Let $\pi : V \to V \cup \{\mathrm{NIL}\}$ map each node to its parent, that is, $\pi(v) = u$ if $(v, u) \in E_K$ and $\pi(v) = \mathrm{NIL}$ if there exists no parent. By definition,

$$\begin{aligned}
|\mathcal{T}_K| &= \sum_{v \in V, \pi(v) \ne \mathrm{NIL}} w(v, \pi(v)) + L \cdot |\{v : \pi(v) = \mathrm{NIL}\}| \\
&\ge \sum_{v \in V, \pi(v) \ne \mathrm{NIL}} \mathrm{rep}(v) + \sum_{v \in V, \pi(v) = \mathrm{NIL}} \mathrm{rep}(v) \\
&= \sum_{v \in V} \mathrm{rep}(v).
\end{aligned}$$

With the above lemma, we can see the greedy algorithm indeed returns a read forest with the minimum total weight (i.e., number of symbols) since it guarantees for any node $v$, the parent it will identify, $\pi(v)$, will satisfy $w(v, \pi(v)) = \mathrm{rep}(v)$; thus, we have the following theorem.

**Theorem 1** *The greedy algorithm computes $\mathcal{T}_K^*$.*

**An information theoretic lower bound for HTS data compression**. In the previous section, we established that Assembltrie constructs the smallest

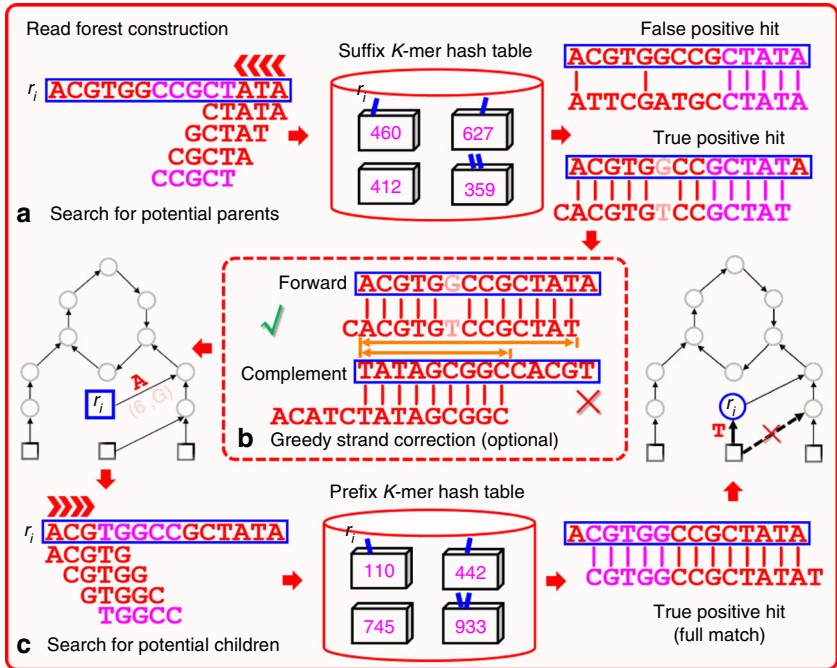

**Fig. 5** The read forest construction process of Assembltrie. To place each read in a cycle-rooted trie, Assembltrie greedily identifies its parent and children by the use of prefix and suffix $K$-mer hash tables. The initial $K$-mer matches (i.e., hash table hits) are extended allowing a maximum number $\epsilon$ of mismatches. As a user option, the greedy strand synchronization heuristic picks for each read, the strand that has the longest prefix–suffix overlap with the implied parent

representation of error-free reads among all descriptions that take the form of a read forest. However, this does not preclude the possibility that representation of reads by a fundamentally different data structure could obtain better compression performance. In this section, we address this question through the lens of information theory. More specifically, in his seminal 1948 paper[22], Shannon proved that any realization of a random source can, on average, be described by a number of bits not exceeding the entropy of the source. Conversely, any compression scheme that uses fewer bits is doomed to encounter errors in reproducing the source from its compressed form. The key point is that this latter bound on compression performance applies uniformly to any compression algorithm, and therefore provides a fundamental benchmark that no viable compression scheme can outperform. In this section, we attempt to estimate this performance limit for HTS data under realistic assumptions. Later in the paper, we demonstrate that Assembltrie comes close to achieving this bound on bacterial genomes across a spectrum of read-error rates.

For a random variable $X$ taking values in a set $\mathcal{X}$ with probability mass function $p_X$, the entropy associated to $X$ is given by:

$$H(X) := \sum_{x \in \mathcal{X}} p_X(x) \log \frac{1}{p_X(x)},$$

where $\log(\cdot)$ denotes the base-2 logarithm, and $H(X)$ has units of bits. Direct computation of the entropy associated to a set of reads $\mathcal{R}$ sampled from a genome $G$ is not practical since a probability distribution over possible genomes $G$ is not available. Nevertheless, it is well known that universal compression algorithms, such as LZ77, are guaranteed to produce descriptions that approach the entropy of a random source with an unknown probability distribution under mild assumptions[23–26]. As such, we are justified in estimating $H(G)$ by $LZ(G)$, defined as the description length of $G$ produced by the universal compression algorithm LZ77 (as implemented in gzip). Using this, we may approximate $H(\mathcal{R})$, the minimum number of bits needed by any algorithm to describe the reads $\mathcal{R}$. Specifically, we argue below that

$$H(\mathcal{R}) \approx NL \log(3) h_2(p) + |G| \cdot H(\text{Poisson}(N/|G|)) + LZ(G), \quad (1)$$

where $h_2(p) = -p \log(p) - (1-p) \log(1-p)$ is the binary entropy of $p \in (0, 1)$, $H(\text{Poisson}(N/|G|))$ denotes the entropy of a Poisson random variable with mean $N/|G|$, $N$ is the number of reads, and $|G|$ denotes the length of the sequence $G$. This approximation is generally valid under the following assumptions:

- Poisson sampling: Reads in $\mathcal{R}$ are assumed to be sampled independently and uniformly at random from all positions in $G$. In such sampling, the number of reads at each position is well-approximated by independent Poisson(N/|G|) random variables for $N$ and $|G|$ large.
- Independent substitution errors: We assume that each base of each read in $\mathcal{R}$ is

corrupted independently with probability $p \in (0, 1)$. When a particular base is selected for corruption, it is substituted with another base, chosen with equal probability from the remaining three bases.

**Estimating $H(\mathcal{R})$.** Our goal in this section is to justify the approximation in (1) for $H(\mathcal{R})$, the entropy of the set of reads sampled from a genome $G$, possibly containing read errors. To do this, we first make the assumption that the random process of sampling reads from $G$ is independent of the read-error process that corrupts individual bases through substitutions, insertions, or deletions. In other words, if $\mathcal{R}^\star$ denotes the set of error-free reads (i.e., the reads in $\mathcal{R}^\star$ coincide with those in $\mathcal{R}$, but do not contain any read errors), then the genome $G$ and the sampled reads (with errors) $\mathcal{R}$ are conditionally independent given the corresponding error-free reads $\mathcal{R}^\star$.

Using this assumption and elementary properties of entropy[27], we may write

$$H(\mathcal{R}) = \underbrace{H(\mathcal{R}|\mathcal{R}^\star)}_{\text{entropy of error process}} + \underbrace{H(\mathcal{R}^\star|G)}_{\text{entropy of read sampling process}} + \underbrace{H(G)}_{\text{entropy of sequences}}$$
$$- H(G|\mathcal{R}) - H(\mathcal{R}^\star|G, \mathcal{R}),$$

where $H(\cdot \mid \cdot)$ denotes conditional entropy. As denoted, the first three terms in the expression for $H(\mathcal{R})$ have intuitive interpretations, and may be estimated in practice:

- The first term $H(\mathcal{R}|\mathcal{R}^\star)$ is the entropy of the reads $\mathcal{R}$, given that the error-free reads $\mathcal{R}^\star$ are available to us. That is, $H(\mathcal{R}|\mathcal{R}^\star)$ is roughly equal to the number of bits needed to describe the read errors, if the error-free reads themselves were already known. For example, if the read-error process corresponds to corrupting each base independently with probability $p \in (0, 1)$ by replacing it randomly with one of the remaining three bases, we have $H(\mathcal{R}|\mathcal{R}^\star) = NL \log(3) h_2(p)$, where $N$ denotes the total number of reads, and $L$ denotes their length (in bases).
- The second term $H(\mathcal{R}^\star|G)$ denotes the entropy of the error-free reads, given that the sequence is known. In other words, this is the entropy of the random process of sampling reads from different locations in the genome. If we adopt the Poisson sampling model, in which reads are sampled uniformly at random from the genome, we have to first order the approximation

$$H(\mathcal{R}^\star|G) \approx |G| \cdot H(\text{Poisson}(N/|G|)),$$

where $H(\text{Poisson}(N/|G|))$ denotes again the entropy of a Poisson random variable with mean $N/|G|$, where $N$ is the number of reads, and $|G|$ denotes the length of the sequence $G$. Indeed, given

that the sequence $G$ is known, the entropy of the reads is determined entirely by the locations of the reads on the genome $G$. For $N$ and $|G|$ both modestly large, the number of reads sampled from each locus are well approximated as independent Poisson random variables, each with mean $N/|G|$, provided the number of non-unique $L$-mers in the genome is much smaller than $|G|$.

- The third term $H(G)$, as already discussed above, corresponds to the descriptive complexity of the sequence $G$ itself and may thus be estimated by $LZ(G)$, the description length of $G$ produced by the universal compressor LZ77 (implemented by gzip).

In practice, the contribution of the remaining terms $H(G|\mathcal{R})$ and $H(\mathcal{R}^\star|G, \mathcal{R})$ appearing in the decomposition of $H(\mathcal{R})$ will be negligible compared to the first three terms described above. To be more precise, the term $H(G|\mathcal{R})$ represents our remaining uncertainty about the sequence $G$ (in bits), given that we observe the sampled reads $\mathcal{R}$. In practice, the set of reads $\mathcal{R}$ is generally rich enough (in terms of coverage) to permit reliable assembly of a small number of contigs, which collectively cover the sequence $G$. Thus, $H(G|\mathcal{R})$ is at most the logarithm of the number of arrangements of the assembled contigs, so that

$$H(G|\mathcal{R}) \leq \log(N_c!) \leq N_c \log N_c,$$

where $N_c$ denotes the number of contigs assembled from $\mathcal{R}$. Since $N_c$ will generally be several orders of magnitude smaller than $NL$ or $|G|$, the contribution of $H(G|\mathcal{R})$ to computing $H(\mathcal{R})$ will be negligible in practical settings. In the case of error-free reads, this heuristic argument can be made rigorous, in which case $H(G|\mathcal{R})$ is only on the order of tens of bits for real genomes[28].

Finally, the term $H(\mathcal{R}^\star|G, \mathcal{R})$ is seen to be small as follows: the entropy of the error-free reads $\mathcal{R}^\star$ given the sequence $G$ coincides precisely with the entropy of the random locations from which the reads are sampled. However, if the error rate is not too severe, access to both $G$ and $\mathcal{R}$ will effectively determine these locations. Indeed, mapping the reads in $\mathcal{R}$ to the sequence $G$ should, in all but few extreme cases, reveal the locations on the genome from which the reads in $\mathcal{R}^\star$ are sampled. Thus, we may conclude that $H(\mathcal{R}^\star|G, \mathcal{R})$ is much smaller in comparison to $H(\mathcal{R}^\star|G)$, implying that its contribution to the calculation of $H(\mathcal{R})$ can be safely ignored.

Putting everything together yields the approximation (1).

**Implementation of the Assembltrie**. An actual implementation of Assembltrie approach needs to address a number of issues such as read errors, strand correction for diploid or multiploid genomes, and parallelization as discussed below.

Noisy reads: Assembltrie allows a maximum of (user specified) $\varepsilon$ mismatches between a parent and a child in the read forest. For that, for each suffix/prefix hash table hit (of the current window of size $K$) encountered during the search for the parent/children of a read, Assembltrie extends the corresponding initial match to a full match with maximum Hamming distance $\varepsilon$. The encoding of mismatching symbols is described below.

Strand correction: As the underlying DNA sequence is double stranded, and the reads are obtained from either of these two strands, the following greedy heuristic is implemented as a user option to correct (i.e., synchronize) the read orientation: for each node $v$, pick the strand (i.e., either the read as is or its reverse complement) for which the prefix–suffix overlap between $v$ and an already processed read $u$ is as long as possible. Then $\pi(v)$ is set to $u$ with the strand of $v$. Note that when $\pi(v)$ is reset afterward, the strand of $v$ will not change.

Trie encoding: Assembltrie encodes the read forest $\mathcal{T}$ in a simple, bottom-up fashion, which preserves the total weight of $\mathcal{T}$. To be specific, the encoding is done by breaking down the constructed read forest back into a set $\mathcal{S}$ of disjoint contigs, each of which represents a simple path from a leaf node to an internal node or root, with its own metadata indicating the start indices of the reads it includes. Each step of the encoding process starts with an unprocessed leaf node (the special case of single cycles is detected and handled afterward), traversing the path to the root, until reaching an already processed node, or otherwise the root itself. The contig $S \in \mathcal{S}$ corresponding to such a path is represented by the Assembltrie encoder as a 6-tuple:

- $S' = \pi(S)$: a pointer to the parent sequence $S'$ of $S$, or NIL if the last read in $S$ corresponds to a root node. Note that it is possible to have $S' = S$, which implies that a single cycle exists in $S$.
- $\{start(r)\}$: the start location of each read $r$ contained in $S$, encoded differentially with a fixed-codebook Huffman code. The sum of the differential codes is $|seq(S)|$ (i.e., the number of symbols to be read in a decompression process from the main stream—as defined below).
- $\{rev(r)\}$: binary flags indicating whether each read $r$ is in its original order, or its reverse-complement order. (This only takes effect if the read-orientation correction heuristic is applied by the user.)
- $r' \in S'$: a pointer to a parent read of $S$ encoded with $\log_2|n(S')|$ bits, where $n(S')$ is the number of reads in contig $S'$, provided that $\pi(S) \neq$ NIL.
- $seq(S)$: the main stream of symbols indicating the consensus sequence of $S$ (an

example is given in Fig. 4b). Although each base can be naively encoded in 2 bits, (note that the letter "N" is provisionally regarded as an arbitrary symbol in $\{A, C, G, T\}$) in Assembltrie, the main stream may be further compressed with a universal compressor.
- $\{error(r)\}$: a record of all mismatching symbols, as well as their positions, between each read $r$ and the consensus sequence $seq(S)$.

In addition to the above encoding, a separate data stream is maintained for the reads for which neither a parent nor a child could be identified (due to insufficient overlaps in the filtered read overlap graph $G_K$). Finally, a third stream is maintained for representing the positions of symbol N. Note that even though these positions are detectable through the use of quality scores, Assembltrie is designed to processes only the reads and not the quality scores and thus in order to achieve lossless compression, it needs to explicitly maintain these positions.

Parallelization: Unlike most available HTS compression software, Assembltrie's parallelization is based on a shared memory scheme, which avoids any tradeoff between speed and compression rate. In particular, Assembltrie utilizes the concurrent containers offered by Intel® Threading Building Blocks to construct hash tables that support concurrent insertion and lookup, without explicit locks.

Paired-end reads: As per many available reference-free sequence compressors (including Orcom), Assembltrie is primarily designed for single-end reads. For data sets are comprised of paired-end reads, it is possible to convert each paired-end read to a single-end read by simply concatenating the two ends into a single string. This conversion (which could be applied as a preprocessing step to all reads) works well especially if the insert size (the distance between between the two ends) has limited variation across the reads. Additional ideas for handling paired-end reads are discussed in, for instance[29].

**Data availability**. The source code, including some python scripts used for benchmarking, is available at https://github.com/kyzhu/assembltrie.

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

## Acknowledgements

D.T., T.C., and S.C.S. acknowledge the Algorithmic Challenges in Genomics program at the Simons Institute of Theoretical Computer Science in U.C. Berkeley. This work was supported in part by the National Science Foundation (NSF) grants CCF-1528132 and CCF-0939370 (Center for Science of Information) to T.C.; National Science Foundation (NSF) grant CCF-1619081, National Institutes of Health (NIH) grant GM108348, NSERC Discovery Frontiers Program on the Cancer Genome Collaboratory, and the Indiana University Grant Challenges Program, Precision Health Initiative, to S.C.S.

## Author contributions

J.H. and T.A.C. initially came up with the cycle-rooted trie data structure. K.Z. and S.C.S. formulated the combinatorial problem. A.A.G. and T.A.C. derived the information theoretic lower bound. A.A.G., K.Z., and I.N. implemented the construction of read forest (cycle-rooted tries). A.A.G., T.A.C., K.Z., S.C.S., and D.N.T. co-wrote the manuscript. T.A.C., S.C.S., and D.N.T. supervised the study.

## Additional information

**Competing interests:** The authors declare no competing financial interests.

