## [Peer Review File · Nature Communications]

Reviewers' comments:

Reviewer #1 (Remarks to the Author):

The manuscript by Ginart et al addresses an important problem in sequencing/genomics, that is the compression of DNA sequences generated by second-generation instruments (e.g., Illumina). The authors have devised an efficient compressed representation of short reads based on compact tries. The method has combinatorial optimality for error-free reads. The authors show that their method approximates an information theoretical upper bound on the compression rate achievable. Extensive experimental results show that their method improves compression of FASTA files significantly compared to existing (published) DNA compression methods. The software is available on github. The manuscript is well-written. Experimental results are convincing. While I did not check all the proofs step-by-step, I am confident that they are correct.

MAJOR COMMENTS

- The Introduction does not explain why the problem of compressing DNA sequences is important and challenging
- The optimality claim only applies when the reads are error-free; the abstract simply claims that "we prove that our approach achieves the shortest possible output on any input", but there is no mention about the fact that reads must be error-free
- The author say nothing about decompression. Is the time to decompress much faster than compression? What about RAM usage to decompress?
- Please explain the N/A in Table 2 and 3. Why are BEETL, Mince and K-path's results missing for three datasets? How comes Assembltrie did not work on NA12878?
- Table 3 is not very clear. Does "(t)" mean time? Does "(m)" mean memory? Are you sure that the number in the two columns "(m)" are in GB as you write in the caption? Are you really using tens-hundreds of TB of RAM? Assuming that the numbers are MB, Assembltrie uses up to 470GB of RAM -- would that limit its use in practice? Orcom uses a LOT less RAM. Please address this issue in your manuscript. Also it seems that you had to downsample some datasets in order to produce the results (see caption of Table 1). Please explain why? Was the 1TB of RAM on your server not sufficient?
- On Page 3, the notation $S[i:j]$ to extract a substring from S is not defined
- I do not think that explaining the algorithm that builds the tries in the abstract is appropriate; in addition, the abstract is way too long
- on Page 9, what do you mean by "letter N is provisionally regarded as an arbitrary symbol in $\{A,C,G,T\}$ "? are you saying that if the input reads have Ns, they will be replaced by a random nucleotide? That's not lossless compression.

MINOR COMMENTS

- The latin name for ALL organisms need to be in italics (e.g. *H. sapiens*, *S.cerevisiae*, *T.cacao*, etc.)

Reviewer #2 (Remarks to the Author):

In this manuscript, Ginart et al. present a reference-free compression algorithm for HTS data based on the idea of performing what is essentially a lightweight de novo pre-assembly of reads by building a forest of connected tries. The manuscript also provides a theoretical lower bound on the compression rate. The manuscript presents a novel concept for representing raw read sequences and the results produced by the implemented tool are impressive. I do have some concerns with the current presentation that I believe should be addressed.

Minor Points:

- The GitHub link provided in the abstract of the manuscript is not open to public, it asks for a password to access the repository. However, it seems changing .iu.edu to .com leads to a public GitHub repository.
- The caption of Table 3 seems to be incorrect, as it reports the size of memory usage in GB. While the machine used is reported to have 1TB of RAM, the memory usages reported are much more than that. For example, even on the smallest dataset, Assemblytrie's memory usage is listed as 1372.8 (which would be ~ 1.3 TB if the number is listed in units of GB).

Major Points:

- "Since the reads form a multiset, and because the read locations are arbitrary, we do not maintain the order of the reads." To me, it is not clear from the manuscript how or if assemblytrie handles paired-end reads. Since the permutation order is not stored anywhere, if it compresses each of the ends separately, then the coordination between the left and right end of the read would be lost, defeating the purpose of a paired end sequencing experiment. This is a major concern that should be addressed. Or, if it is already handled by the method, it should be clearly explained how this is done. In schemes, like assemblytrie, where reordering of sequences is allowed, it is not trivial to maintain synchronization between read pairs (and it is not correct to compress the ends of a paired-end fragment independently). One approach is to concatenate both ends of the paired-end read and then compress the resulting strings. However, given the normal variation in the fragment lengths from which the reads were sampled, this generally results in "worse" compression than compressing the files for each end individually. Some of the datasets in the presented benchmark are such paired end files, eg. ERR174310. If it is not already the case, the benchmarks should be redone in a way that properly handles paired-end reads. If it is already the case, the scheme used to handle read-pairs should be explained in detail in the manuscript.
- Similarly, assemblytrie encodes neither read names, read comments or (if the input is a fastq file) read qualities. While I fully believe that sequence compression is an interesting and useful problem to tackle in its own right, at least some space in the manuscript should be dedicated to describing how quality scores and read names might also be retained. It is possible, during the encoding process, to recall the order of the current read in the original file and to write the corresponding quality string to a different stream (e.g. to be encoded by a different mechanism)? Presumably, a naive implementation of this would require storing even more information per-read during the initial encoding step. The authors should discuss such considerations, at least briefly, in the manuscript.
- The read forest or a collection of tries is constructed iteratively. The description of the iterative process is simple, but the clarity and succinctness could be improved. For example "It then greedily picks for v , an existing node as its parent $\pi(v)$, such that the overlap between a prefix of $st(v)$ and a suffix of $st(\pi(v))$ is maximum possible (see below for details), and is of length at least K .", this might be rephrased as "It then greedily picks an existing node $\pi(v)$, satisfying two conditions, i. the overlap between suffix of $\pi(v)$ and prefix of v is maximal among all existing nodes, ii. the overlap size is at least K ". More details can still be given below, but the precise conditions are now stated clearly where the idea is introduced. There are a few more such sentences which should be rewritten.
- In the description of problem statement it is written that each node can only have one outgoing edge, given that each node has only one parent, which is implied from the maximality. I assume ties are broken randomly? Nevertheless, it is better to mention this explicitly, as one could potentially imagine different heuristics (e.g. if there are ties, add the read to what is currently the largest trie etc.).

- The notion of allowing mismatches has been mentioned for the first time in Figure 3. Neither the description nor the analysis before that mention the idea of approximate matching. I think it is worth mentioning that the implementation of the compression algorithm, is based on approximate matching rather than exact matches. It is also worth mentioning what effect, if any, this could have on the analysis (e.g. the trade-off between choosing potentially longer overlaps versus overlaps with fewer mismatches, and what if the total number of shared symbols is the same?).

- In the analysis of running time of the algorithm, the big O for running time is approximated as $O(NL)$, where N is the number of reads and L is the length of individual read string. The analysis for insertion of a new read is, I think, worth investigating. For example, genomes are often not actually sampled uniformly at random (at least with short reads), and some regions might be considerably over or undersampled. Also, as the authors mention, long genomic repeats can lead to highly over-represented sequence (k-mer) content. To emphasize the effect of "real data", it would be useful to present a figure that shows both the theoretical and empirical timing as a function of N (and potentially also L).

- There is almost no mention about how the decompression process works. It is also not explained in detail how the mismatches along with the position for the reads are stored — there should be some text (perhaps as supplementary material) describing the exact storage/encoding scheme that is implemented etc. Overall the method description is incomplete without full details about the implemented. It would also be very useful to provide pseudocode for the encoding / decoding algorithms, explaining the working principle of compression and decompression.

- The time and memory benchmarks in the paper cover only the compression / encoding state. However, since, for practical usage, one is typically interested in fast and efficient decompression, time and memory benchmarks should also be provided for the decompression stage of the algorithms that are compared.

- Finally, I would suggest including, at least for a point of basic comparison, the performance of at least one reference-based tool — this might provide an idea of the relative efficiency of encoding with respect to a "light assembly" versus a standard reference.

Overall the manuscript is well-written and presents a novel concept of light assembly which can be extended in many useful ways.

Dear Editor,

Below, please find our detailed response to the reviewers' comments and questions on our manuscript entitled "Optimal Compressed Representation of High Throughput Sequence Data via Light Assembly" which is currently in consideration to be published in Nature Communications. We believe we have addressed all of the reviewers' concerns in our significantly revised manuscript.

Reviewer #1 (Remarks to the Author):

The manuscript by Ginart et al addresses an important problem in sequencing/genomics, that is the compression of DNA sequences generated by second-generation instruments (e.g., Illumina). The authors have devised an efficient compressed representation of short reads based on compact tries. The method has combinatorial optimality for error-free reads. The authors show that their method approximates an information theoretical upper bound on the compression rate achievable. Extensive experimental results show that their method improves compression of FASTA files significantly compared to existing (published) DNA compression methods. The software is available on github. The manuscript is well-written. Experimental results are convincing. While I did not check all the proofs step-by-step, I am confident that they are correct.

MAJOR COMMENTS

- The Introduction does not explain why the problem of compressing DNA sequences is important and challenging

Please see the first paragraph in the Introduction of the revised manuscript.

- The optimality claim only applies when the reads are error-free; the abstract simply claims that "we prove that our approach achieves the shortest possible output on any input", but there is no mention about the fact that reads must be error-free

We needed to shorten the abstract to satisfy the Nature Communications guidelines but explain in the introduction that our optimality claims hold for error free reads only.

- The author say nothing about decompression. Is the time to decompress much faster than compression? What about RAM usage to decompress?

We have added a new table to present the decompression times and memory usage. As we can see, compared with tools that depend on various down-stream encoders like Orcom, Assembltrie requires shorter decompression time, with higher space requirements - \$\Omega(NL)\$ where N is the number of reads and L is read length - for lossless decompression.

- Please explain the N/A in Table 2 and 3. Why are BEETL, Mince and K-path's results missing for three datasets? How comes Assembltrie did not work on NA12878?

Those (particularly large) datasets are not part of the MPEG HTS benchmarking data collection (this has been clarified in the table now) and thus the performance figures of any of the tools we tested were not available in [Numanagic et al 2016]. We only report the performance figures of Assembltrie and Orcom on these data sets since BEETL, Mince and K-path were very slow and achieved compression rates much worse than Orcom on MPEG HTS data set [Numanagic et al 2016].

We do report on Assembltrie's performance on NA12878 with no strand correction - however, as the reviewer observed, we do not report on its performance with strand correction. This is because reads in this data set are "already strand corrected" (they are extracted from a BAM file) - i.e. Assembltrie's performance is the same with or without strand correction.

- Table 3 is not very clear. Does "(t)" mean time? Does "(m)" mean memory? Are you sure that the number in the two columns "(m)" are in GB as you write in the caption? Are you really using tens-hundreds of TB of RAM? Assuming that the numbers are MB, Assembltrie uses up to 470GB of RAM -- would that limit its use in practice? Orcom uses a LOT less RAM. Please address this issue in your manuscript. Also it seems that you had to downsample some datasets in order to produce the results (see caption of Table 1). Please explain why? Was the 1TB of RAM on your server not sufficient?

We thank the reviewer for noticing this mistake - the numbers are indeed in MBs. Table 3 is now revised to make it clearer. As the reviewer observes, Orcom indeed needs less space; Assembltrie's current implementation requires $\Theta(NL)$ memory (the actual memory usage is roughly 3 times the input file size). It may be possible to reduce its memory usage with more advanced data structures but that may result in a slow down. Our main goal in this paper is to improve compression while maintaining reasonable speed so we leave this to future work.

The reason for downsampling ERP001775 dataset is that it is a very large data set combining reads from 18 human individuals. Such datasets are not commonly encountered in practice. We only included this dataset for completeness. We did NOT downsample NA12878 "to produce results". Rather (1) this data set is NOT a part of the MPEG HTS FASTQ/FASTA benchmarking data set - it is rather a BAM file from which we extracted individual reads - which are, by definition, strand corrected. (2) We downsampled it to match the coverage of ERR174310 for the purpose of demonstrating the comparative performance of Assembltrie's strand correction heuristic. This gave us two similar coverage human genome data with identical read lengths, one strand corrected by our heuristic, the other strand corrected by mapping. We demonstrated that while our strand correction heuristic is good, it has a long way to match the strand correction achievable by mapping. Note that we used low coverage samples for this experiment because the performance difference is the biggest in low coverage data.

- On Page 3, the notation $S[i:j]$ to extract a substring from S is not defined

We have included a definition in the revised manuscript.

- I do not think that explaining the algorithm that builds the tries in the abstract is appropriate; in addition, the abstract is way too long

We have shortened the abstract to satisfy Nature Communications format requirements. Please check the revised manuscript.

- on Page 9, what do you mean by "letter N is provisionally regarded as an arbitrary symbol in {A,C,G,T}"? are you saying that if the input reads have Ns, they will be replaced by a random nucleotide? That's not lossless compression.

Lossless compression is achieved by keeping the positions of all symbols N and replacing each N with an arbitrary symbol from {A, C, G, T}. Our compressed representation includes these positions and hence is lossless.

MINOR COMMENTS

- The latin name for ALL organisms need to be in italics (e.g. *H. sapiens*, *S.cerevisiae*, *T.cacao*, etc.)

Thanks for pointing out the issue and it has been corrected.

Reviewer #2 (Remarks to the Author):

In this manuscript, Ginart et al. present a reference-free compression algorithm for HTS data based on the idea of performing what is essentially a lightweight de novo pre-assembly of reads by building a forest of connected tries. The manuscript also provides a theoretical lower bound on the compression rate. The manuscript presents a novel concept for representing raw read sequences and the results produced by the implemented tool are impressive. I do have some concerns with the current presentation that I believe should be addressed.

Minor Points:

- The GitHub link provided in the abstract of the manuscript is not open to public, it asks for a password to access the repository. However, it seems changing [.iu.edu](https://github.com/kyzhu/assemblytrie) to [.com](https://github.com/kyzhu/assemblytrie) leads to a public GitHub repository.

The source code, including some python scripts used for benchmarking is available at <https://github.com/kyzhu/assemblytrie>. Datasets are available at <ftp://ftp.sra.ebi.ac.uk/vol1/fastq> and elsewhere. Reference genomes are available at <https://www.ncbi.nlm.nih.gov/>.

- The caption of Table 3 seems to be incorrect, as it reports the size of memory usage in GB. While the machine used is reported to have 1TB of RAM, the memory usages reported are much more than that. For example, even on the smallest dataset, Assemblytrie's memory usage is listed as 1372.8 (which would be ~1.3TB if the number is listed in units of GB).

We thank the reviewer for noticing this mistake - which we have corrected. The memory usage is indeed in MBs.

Major Points:

- "Since the reads form a multiset, and because the read locations are arbitrary, we do not maintain the order of the reads." To me, it is not clear from the manuscript how or if Assemblytrie handles paired-end reads. Since the permutation order is not stored anywhere, if it compresses each of the ends separately, then the coordination between the left and right end of the read would be lost, defeating the purpose of a paired end sequencing experiment. This is a major concern that should be addressed. Or, if it is already handled by the method, it should be clearly explained how this is done. In schemes, like Assemblytrie, where reordering of sequences is allowed, it is not trivial to maintain synchronization between read pairs (and it is not correct to compress the ends of a paired-end fragment independently). One approach is to concatenate both ends of the paired-end read and then compress the resulting strings. However, given the normal variation in the fragment lengths from which the reads were sampled, this generally results in "worse" compression than compressing the files for each end individually. Some of the datasets in the presented benchmark are such paired end files, eg. ERR174310. If it is not already the case, the benchmarks should be redone in a way that properly handles paired-end reads. If it is already the case, the scheme used to handle read-pairs should be explained in detail in the manuscript.

We thank the reviewer for bringing this issue to our attention. We note that the best FASTQ compressors such as Orcom do not truly support paired-end read compression. Orcom compressed files, as the reviewer has pointed out, do not maintain synchronization between read pairs. In addition, our main focus in this paper are single end reads.

Nevertheless we have tested Assemblytrie and Orcom on the MPEG HTS benchmarking dataset (SRR554369) after concatenating paired end reads into single (longer, i.e. 200bp) reads. This provides lossless compression. As can be seen below Assemblytrie improves on Orcom on this dataset as well.

Dataset	Assemblytrie	Orcom
SRR554369: paired-end reads concatenated (L = 200)	53.97M	70.44M

- Similarly, assembltrie encodes neither read names, read comments or (if the input is a fastq file) read qualities. While I fully believe that sequence compression is an interesting and useful problem to tackle in its own right, at least some space in the manuscript should be dedicated to describing how quality scores and read names might also be retained. It is possible, during the encoding process, to recall the order of the current read in the original file and to write the corresponding quality string to a different stream (e.g. to be encoded by a different mechanism)? Presumably, a naive implementation of this would require storing even more information per-read during the initial encoding step. The authors should discuss such considerations, at least briefly, in the manuscript.

Newly available FASTQ compressors such as Mince and Orcom compress only the sequence content of the reads - and not the read names or quality scores. One key reason is that the encoding of read names and quality scores provided by SCALCE [Hach et al 2012] - which was developed by some of us - and others perform quite well as demonstrated in [Numanagic et al 2016]. The redundancy in these fields have different characteristics in comparison to the redundancy in the sequence information - whose reduction is the main focus of our study.

- The read forest or a collection of tries is constructed iteratively. The description of the iterative process is simple, but the clarity and succinctness could be improved. For example "It then greedily picks for v , an existing node as its parent $\pi(v)$, such that the overlap between a prefix of $st(v)$ and a suffix of $st(\pi(v))$ is maximum possible (see below for details), and is of length at least K .", this might be rephrased as "It then greedily picks an existing node $\pi(v)$, satisfying two conditions, i. the overlap between suffix of $\pi(v)$ and prefix of v is maximal among all existing nodes, ii. the overlap size is at least K ". More details can still be given below, but the precise conditions are now stated clearly where the idea is introduced. There are a few more such sentences which should be rewritten.

We thank the reviewer for this suggestion; we modified the above description accordingly.

- In the description of problem statement it is written that each node can only have one outgoing edge, given that each node has only one parent, which is implied from the maximality. I assume ties are broken randomly? Nevertheless, it is better to mention this explicitly, as one could potentially imagine different heuristics (e.g. if there are ties, add the read to what is currently the largest trie etc.).

The reviewer is correct that these ties are broken arbitrarily but this will have no effect on the performance since no heuristic can change the number of symbols in the read forest (due to proven optimality), or will have an effect on the information theoretic bound (which is algorithm-independent).

- The notion of allowing mismatches has been mentioned for the first time in Figure 3. Neither the description nor the analysis before that mention the idea of approximate matching. I think it

is worth mentioning that the implementation of the compression algorithm, is based on approximate matching rather than exact matches. It is also worth mentioning what effect, if any, this could have on the analysis (e.g. the trade-off between choosing potentially longer overlaps versus overlaps with fewer mismatches, and what if the total number of shared symbols is the same?).

We now mention that the implementation of the algorithm performs approximate matching allowing mismatches. We also mention in the introduction that the optimality claims only apply to error/noise free reads, for which approximate matching is not necessary. The effect of noise is investigated in the context of information theoretic bounds. However we believe an analysis for the tradeoff between longer overlaps and resulting compression depends highly on the noise model and thus is beyond the scope of our paper.

- In the analysis of running time of the algorithm, the big O for running time is approximated as $O(NL)$, where N is the number of reads and L is the length of individual read string. The analysis for insertion of a new read is, I think, worth investigating. For example, genomes are often not actually sampled uniformly at random (at least with short reads), and some regions might be considerably over or undersampled. Also, as the authors mention, long genomic repeats can lead to highly over-represented sequence (k-mer) content. To emphasize the effect of "real data", it would be useful to present a figure that shows both the theoretical and empirical timing as a function of N (and potentially also L).

This is indeed a good point. In order to see the effect of sampling in real data vs uniformly sampled reads, with varying values of N (the number of reads) please check the table below. As can be seen Assembltrie is robust with respect to the sampling bias.

Dataset: E. coli DH10B	Real (potentially bias-sampled)	Uniformly Sampled
L = 150, cov = 30	21.2s	21.1s
L = 150, cov = 35	19.5s	19.2s
L = 150, cov = 40	15.8s	15.9s

- There is almost no mention about how the decompression process works. It is also not explained in detail how the mismatches along with the position for the reads are stored — there should be some text (perhaps as supplementary material) describing the exact storage/encoding scheme that is implemented etc. Overall the method description is incomplete without full details about the implemented. It would also be very useful to provide pseudocode for the encoding / decoding algorithms, explaining the working principle of compression and decompression.

A detailed description of decompression can be found in section “*Implementation of the Assembltrie*”. Note that mismatch positions are encoded differentially - together with substitution symbols for each read. We believe that a pseudocode will simply confuse the reader as it would be unnecessarily long. In addition, our code is open source.

- The time and memory benchmarks in the paper cover only the compression / encoding state. However, since, for practical usage, one is typically interested in fast and efficient decompression, time and memory benchmarks should also be provided for the decompression stage of the algorithms that are compared.

Assembltrie decompression time and memory in comparison to that of Orcom are now presented in Table 4.

- Finally, I would suggest including, at least for a point of basic comparison, the performance of at least one reference-based tool — this might provide an idea of the relative efficiency of encoding with respect to a "light assembly" versus a standard reference.

Please see Table 2 (compression benchmarking table); here k-Path is a reference based tool. Our main goal is to improve on reference free compressors and thus we had not included SAM/BAM compressors in our benchmarking exercise. This is because (1) the running times of such tools are particularly high due to their need to perform read mapping first - a very costly procedure, and (2) a reference genome is not always available.

Nevertheless, following up on the reviewer’s suggestion, we have compared Assembltrie with the best (according to [Numanagic et al 2016]) reference-based SAM/BAM compressor DeeZ.

Dataset	SRR554369		SRR327342	
Index (BWA)	3.9s		9.0s	
Alignment (BWA)	9.8s		110.6s	
Conversion to sam (BWA)	8.0s		117.0s	
Sorting (BWA)	7.3s		94.8s	
Encoding (DeeZ)	15.2s	30.91MB	95.1s	32.16MB
Total-time (BWA+DeeZ)	44.2s		426.5s	
Assembltrie	30.6s	7.11MB	256.2s	29.77MB

As can be seen above, the total running time (mapping and compression) for the best reference based method available is higher than that of Assembltrie (the larger the data set, the bigger the difference). In addition, the compression performance of Assembltrie is much better than that of the reference based method (much better in the smaller data set).

Overall the manuscript is well-written and presents a novel concept of light assembly which can be extended in many useful ways.

References

[Numanagic et al 2016] Numanagić, I., Bonfield, J. K., Hach, F., Voges, J., Ostermann, J., Alberti, C., ... & Sahinalp, S. C. (2016). Comparison of high-throughput sequencing data compression tools. *Nature Methods*, 13(12), 1005-1008.

[Hach et al 2012] Hach, F., Numanagić, I., Alkan, C., & Sahinalp, S. C. (2012). SCALCE: boosting sequence compression algorithms using locally consistent encoding. *Bioinformatics*, 28(23), 3051-3057.

REVIEWERS' COMMENTS:

Reviewer #1 (Remarks to the Author):

In this revised manuscript, the authors have satisfactorily addressed all my comments and questions.

Reviewer #2 (Remarks to the Author):

The authors have generally addressed all of my concerns in their revision. However, I would like to suggest that the reviewers include at least some text in the actual manuscript describing how Assembltrie is focused on the problem of compressing single-end reads. Currently, that relevant text appears in the response to the reviewers, but not in the manuscript. The reason I think this is important is because there are different (potentially orthogonal) challenges in how to best compress paired-end data (see, e.g. some of the ideas suggested in Pritt and Langmead's Boiler [1]). This remains an active and important challenge given the prevalence of paired-end protocols in current sequencing experiments.

References:

[1] Jacob Pritt, Ben Langmead; Boiler: lossy compression of RNA-seq alignments using coverage vectors, *Nucleic Acids Research*, Volume 44, Issue 16, 19 September 2016, Pages e133, <https://doi.org/10.1093/nar/gkw540>

Dear Editor,

Please find our final response to the reviewers' comments on our manuscript entitled "Optimal Compressed Representation of High Throughput Sequence Data via Light Assembly" which is currently in the process to be published in Nature Communications. Following the reviewer's suggestion, we have added a new paragraph in the revised manuscript.

Reviewer #2 (Remarks to the Author):

The authors have generally addressed all of my concerns in their revision. However, I would like to suggest that the reviewers include at least some text in the actual manuscript describing how Assembltrie is focused on the problem of compressing single-end reads. Currently, that relevant text appears in the response to the reviewers, but not in the manuscript. The reason I think this is important is because there are different (potentially orthogonal) challenges in how to best compress paired-end data (see, e.g. some of the ideas suggested in Pritt and Langmead's Boiler [1]). This remains an active and important challenge given the prevalence of paired-end protocols in current sequencing experiments.

References:

[1] Jacob Pritt, Ben Langmead; Boiler: lossy compression of RNA-seq alignments using coverage vectors, Nucleic Acids Research, Volume 44, Issue 16, 19 September 2016, Pages e133, <https://doi.org/10.1093/nar/gkw540>

We have added the following paragraph in the revised manuscript:

As per many available reference-free sequence compressors (including Orom) Assembltrie is primarily designed for single-end reads. For data sets are comprised of paired-end reads it is possible to convert each paired-end read to a single-end read by simply concatenating the two ends into a single string. This conversion (which could be applied as a preprocessing step to all reads) works well especially if the insert size (the distance between between the two ends) has limited variation across the reads. Additional ideas for handling paired-end reads are discussed in [1].